# A control theory approach to optimal pandemic mitigation

**Prakhar Godara**, **Stephan Herminghaus**, **Knut M. Heidemann** *

Max Planck Institute for Dynamics and Self-Organization, Göttingen, Germany

* knut.heidemann@ds.mpg.de

## Abstract

In the framework of homogeneous susceptible-infected-recovered (SIR) models, we use a control theory approach to identify optimal pandemic mitigation strategies. We derive rather general conditions for reaching herd immunity while minimizing the costs incurred by the introduction of societal control measures (such as closing schools, social distancing, lockdowns, etc.), under the constraint that the infected fraction of the population does never exceed a certain maximum corresponding to public health system capacity. Optimality is derived and verified by variational and numerical methods for a number of model cost functions. The effects of immune response decay after recovery are taken into account and discussed in terms of the feasibility of strategies based on herd immunity.

## Introduction

The recent outbreak of the illness COVID-19, caused by the SARS-CoV-2 virus, has resulted in a pandemic with unprecedented impact on societies all over the globe. Mitigation measures included complete lockdowns of societal life, with severe psychic, social, and economic consequences [1, 2]. Hence a major focus of governments is on designing containment strategies which are as mild as possible, but substantial enough to limit the severity of the outbreak in order not to overwhelm the health service system (HSS). This requires reliable forecast, based on careful collection of data on the fraction of infected citizens, as well as extensive simulation [2, 3].

We discuss the system in terms of a so-called SIR model [4], referring to the fraction of susceptible (S), infected (I), and recovered (R) citizens in the population. We identify the recovered with all those who are neither susceptible nor infected ($R = 1 - S - I$); the dynamics are thus fully described by a set of two equations:

$$\partial_t S = -\beta SI,$$

$$\partial_t I = \beta SI - \frac{I}{\tau},$$

(1)

where $S, I \in [0, 1]$ are the fraction of susceptible and infected individuals in the population, respectively. Note that $I(t)$ denotes the actual fraction of acutely infected citizens at time $t$, no

**Data Availability Statement:** All data files and code are available on github: https://github.com/poss-group/covid19-control.

**Funding:** P. G., S. H., and K. M. H. gratefully acknowledge support from the Max Planck Society.

**Competing interests:** The authors have declared that no competing interests exist.

matter whether or not the infection has been realized by the individual, or has even been recorded. The infection of a susceptible individual by an infected is described by the *infection rate* $\beta > 0$, while $\tau$ is the *average duration* of the infection of an individual until her recovery.

The task we address in this study is to limit, during the whole period of the pandemic, the current number of infected individuals such as to prevent the number of those needing intensive care from exceeding the capacity of the deployed HSS. Such control may be described by a control parameter $\alpha(t)$, which quantifies the effect of mitigation strategies upon the infection rate. We may write $\beta = \beta_0(1 - \alpha)$, such that $\alpha = 0$ and $\alpha = 1$ correspond to usual societal life and complete mutual isolation of citizens, respectively.

In order to define $\alpha$ in a general way, we state that a certain value of $\alpha = 1 - \beta/\beta_0$ denotes the subset of all possible mitigation measures which lead to an infection rate $\beta \leq \beta_0$. We thus do not need to refer to any specific measures, but can formulate our approach in a very general way. The (more or less) accurate determination of these subsets is then the task of careful social (e.g., infection history) data analysis among citizens. This is illustrated in Fig 1, where mitigation strategies, followed by the public authorities, are indicated by the dashed and dotted curves, within a space spanned by the effect of the measures upon the infection rate, $\alpha$, and the cost incurred for economy and society as a whole, $f(\alpha)$.

As indicated by the explicit time dependence of $\alpha(t)$, we follow a control theoretic approach. This is in some contrast to earlier treatments which have assumed mitigation measures to be constant over time [5–8]. Instead, we aim at determining the optimal function $\alpha(t)$ which minimizes the impact on society, while at the same time avoiding the HSS to become overloaded.

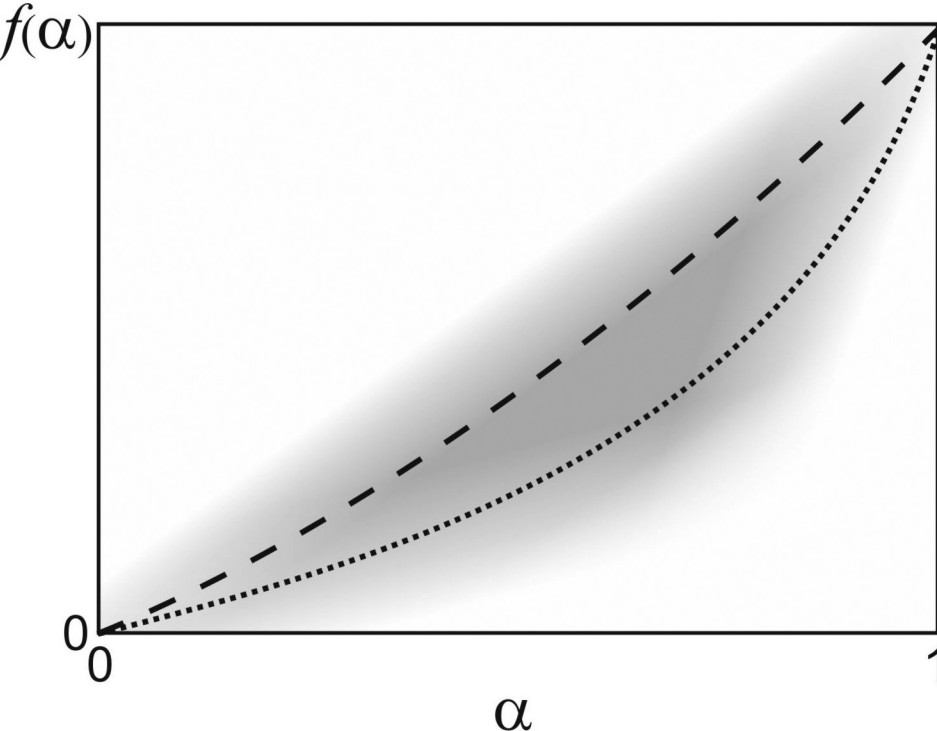

**Fig 1. Space of mitigation measures.** Sketch of the space of possible mitigation measures (grey shade), spanned by their effect on the infection rate, $\alpha$, and their socio-economic cost, $f(\alpha)$. Normal societal life is at the origin, while the upper right corner corresponds to total mutual isolation of all citizens, which is the strongest possible intervention. The dashed and dotted curves depict possible choices for mitigation measures. Such curves correspond to the *cost functions* referred to in the manuscript (see Eq (7).

At the end of the mitigation scenario, *herd immunity* shall be reached, so that the epidemic comes to an end without further control. We do not consider potential vaccination scenarios here (as is done elsewhere [9, 10]), so immunization can only be achieved via infection with the virus in the present study.

A dimensionless quantity which is frequently used in epidemiology is the *reproduction number*, $R := \beta \tau S$, which denotes the average number of susceptibles infected by one infected individual. At the beginning of an epidemic ($S \approx 1$) and without mitigation measures deployed ($\beta = \beta_0$), one observes the *basic* reproduction number $R_0 = \beta_0 \tau$. In case of the COVID-19 pandemic, typical estimates are $R_0 \approx 3$ [11] and $\tau \approx$ ten days [1, 12]. It has been shown before that the inherently random, network-like structure of interactions between individuals mainly results in a readjustment of $R_0$ [13]. Hence we follow a mean field approach, disregarding small scale inhomogeneities of the system. We consider a homogeneous scenario, where $\beta(t)$ depends on time, but is spatially constant. Defining a normalized time variable, $\theta := \beta t$, we can rewrite Eq (1) as

$$\partial_\theta S = -SI \,,$$

(2)

$$\partial_\theta I = SI - \frac{I}{R_0(1-\alpha)} \,.$$

The trajectory of the system in the *S*-*I*-plane, $I(S)$, can be obtained by dividing the equations displayed in Eq (2) by each other. This yields

$$\frac{dI}{dS} = \frac{1}{R_0(1-\alpha)S} - 1 \,,$$

(3)

which has the solution:

$$I(S) = \frac{\ln S}{R_0(1-\alpha)} - S + C \,,$$

(4)

where $C$ is a constant of integration. When no mitigation measures are in place ($\alpha = 0$), we have $I(1) = 0$ and thus obtain

$$I(S) = \frac{\ln S}{R_0} - S + 1 \,.$$

(5)

This is plotted as the dashed curve in Fig 2 for the case $R_0 = 3$. The maximum turns out to occur at $S_{\text{peak}} = 1/R_0$, where it reaches a value of

$$I_{\text{peak}} = 1 - \frac{\ln R_0}{R_0} - \frac{1}{R_0} \,.$$

(6)

At $S = S_{\text{peak}} = 1/R_0$, the population has reached *herd immunity* since from then on the number of infected citizens decreases until zero.

If the disease is serious, one is faced with the problem that with a fraction of $I_{\text{peak}}$ people being infected, the number of those in need of hospitalization or even intense care may exceed the capacity of the HSS. We denote by $I_h < I_{\text{peak}}$ the maximum fraction of infected citizens which can be managed by the HSS. It is limited by infrastructural aspects, such as the availability of staff or the size and areal density of hospitals, and is indicated by the horizontal dotted line in Fig 2. Any substantial overshoot of the dashed curve over the dotted line constitutes a catastrophe, as a major fraction of the population will then not receive proper health care or

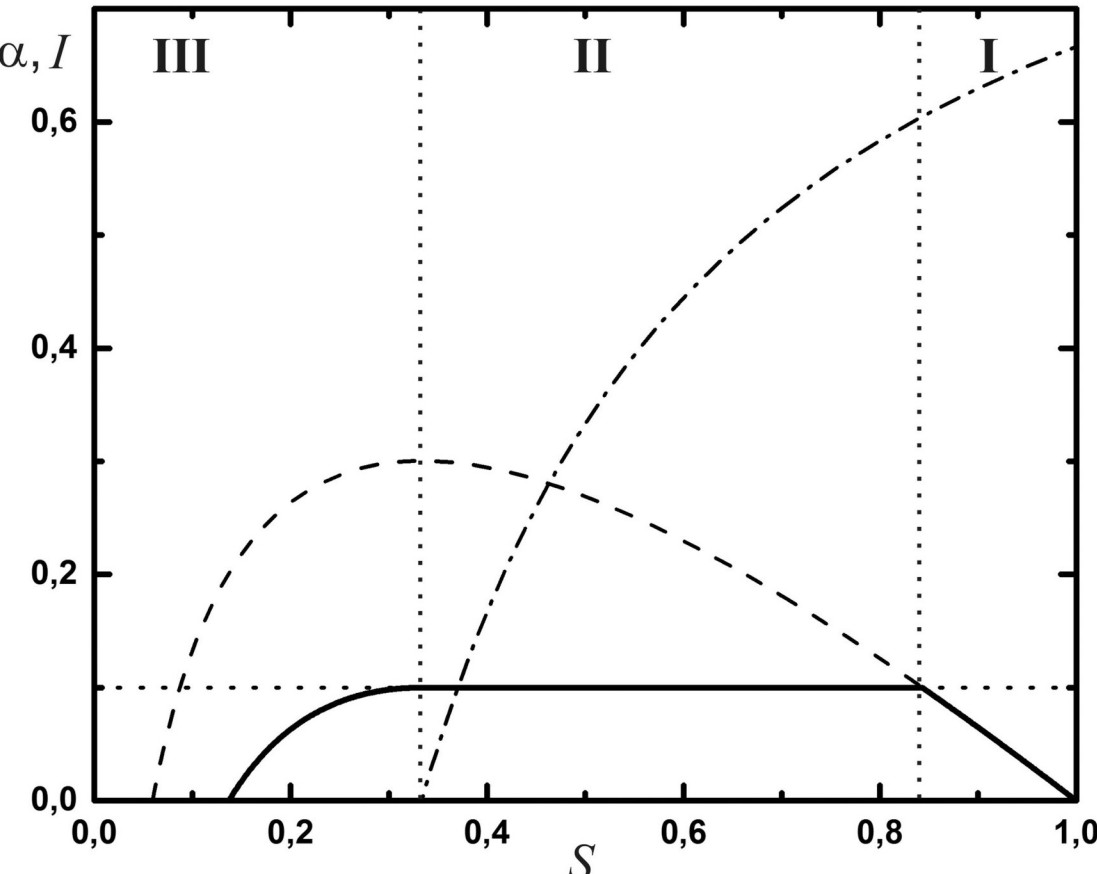

**Fig 2. Trajectories in the (S, I)-plane.** Dashed curve: trajectory with no mitigation starting at $(S, I) = (1, 0)$, $R_0 = 3$. Horizontal dashed line: maximum load of the HSS, $I_h$ (here we have set $I_h = 0.1$ for clarity, although this is unrealistically large). Solid curve: trajectory, $(S^*(t), I^*(t))$, for an optimal choice of $\alpha(t)$ (see Eq 16). The corresponding characteristic of $\alpha(S)$ follows the dash-dotted curve in phase II, the *mitigation phase*. There is no mitigation in phases I and III ($\alpha = 0$).

treatment. This must clearly be avoided by means of suitable measures, such as reducing mutual contacts between individuals, banning major assemblies, reducing mobility etc., thus reducing the infection rate. Such measures are described by the parameter $\alpha(t)$, which is to be discussed next.

## Optimal control problem

It is clear that the aforementioned measures will have a more or less substantial impact on society, mainly through their detrimental effects on economy, but also through other societal (e.g., cultural) damage. This may be described by means of a *cost functional*,

$$K\{\alpha\} := \int f(\alpha(t))\, dt\,, \tag{7}$$

where the cost function $f$ corresponds to the mitigation strategy chosen, i.e., the curve chosen in Fig 1. It denotes the cost incurred at a given control $\alpha$, along with the assumption $\partial f/\partial \alpha \geq 0$ $\forall \alpha$; later we will require $\frac{\partial^2 f}{\partial \alpha^2} \geq 0$. Convexity of the cost function is a reasonable assumption since one can imagine that chosen mitigation measures become more and more costly as $\alpha$ approaches unity. In principle, $f(\cdot)$ could depend on time $t$ explicitly or involve memory, e.g.,

because prolonged measures cause disproportinally higher cost. For simplicity, we do not treat such cases here.

The control problem we choose to address is to find a control trajectory, denoted by the function $\alpha(t)$, such that the impact on society, as described by $K$, is being minimized under the constraint that $I(t)$ never exceeds $I_h$ (capacity of the HSS) and at the end of mitigation—at *unknown* terminal time $t_e$—herd immunity is reached, i.e., $S(t_e) = 1/R_0$ (end of phase II in Fig 2).

We thus need to

$$\text{minimize} \quad K\{\alpha\} = \int_0^{t_e} f(\alpha(t)) \, dt \,,$$

$$\text{such that} \quad \dot{\boldsymbol{x}}(t) = \boldsymbol{h}(\boldsymbol{x}, \alpha(t)) \,, \; \boldsymbol{x}(0) = \boldsymbol{x}_0 \,,$$

$$I(t) \leq I_h \,, \; S(t_e) = 1/R_0 \,, \tag{8}$$

where $\boldsymbol{x} = (S, I)$, and $\boldsymbol{h}(\boldsymbol{x}, \alpha(t))$ as given by Eq 1 (with $\beta = \beta_0(1 - \alpha)$). Minimization of mitigation time is covered by setting $f(\cdot) \equiv const$.

Solving Eq 8 can be recast into minimization of the following functional:

$$J := \int_0^{t_e} f(\alpha(t)) + \boldsymbol{\lambda}(t) \cdot [\dot{\boldsymbol{x}}(t) - \boldsymbol{h}(\boldsymbol{x}, \alpha(t))] + \mu(t)(I - I_h) \, dt \,, \tag{9}$$

where $\boldsymbol{\lambda}(t) = (\lambda_S(t), \lambda_I(t))$, $\mu(t)$ are Lagrange multipliers. The introduction of $\mu(t)$ for the inequality constraint introduces additional constraints on $\mu(t)$, namely $\mu(t) \geq 0$ and the complementary slackness condition $\mu(t)(I^* - I_h) = 0$. These are also known as KKT conditions [14]. The star ($^*$) represents the optimal quantities. Additionally, $S(t_e) = 1/R_0$ and $\boldsymbol{x}(0) = x_0$ need to be enforced.

The necessary conditions for optimality can be evaluated by setting the first variation of Eq 9 to zero (for a detailed derivation see S1 Appendix), we obtain:

$$f(\alpha^*(t_e^*)) - \boldsymbol{\lambda}(t_e^*) \cdot \boldsymbol{h}(\boldsymbol{x}^*(t_e^*), \alpha^*(t_e^*)) = 0 \,, \tag{10}$$

$$\dot{\boldsymbol{\lambda}}(t) = -\boldsymbol{\lambda}(t) \cdot \nabla_x \boldsymbol{h}|_{x^*(t)} + \mu(t) \, \nabla_x (I - I_h)|_{x^*(t)} \,, \tag{11}$$

$$\partial_\alpha f|_{\alpha^*(t)} - \boldsymbol{\lambda}(t) \cdot \partial_\alpha \boldsymbol{h}|_{\alpha^*(t)} = 0 \,, \tag{12}$$

$$\lambda_I(t_e^*) = 0 \,, \tag{13}$$

$$\mu(t) \geq 0 \,, \tag{14}$$

$$\mu(t)(I^* - I_h) = 0 \,. \tag{15}$$

In addition to these, one also has the optimal system dynamics $\dot{\boldsymbol{x}}^*(t) = \boldsymbol{h}(\boldsymbol{x}^*, \alpha^*(t))$. The necessary conditions become sufficient conditions if $\boldsymbol{h}(x, \alpha)$ and $f(\alpha)$ are convex in $x$ and $\alpha$ [15]. The former can be checked to be valid for the SIR model and the latter implies that $\frac{\partial^2 f}{\partial \alpha^2} \geq 0$.

The task remains to find Lagrange multipliers $\boldsymbol{\lambda}(t)$ and $\mu(t)$ which simultaneously satisfy the above conditions. This task usually involves a numerical search for the initial conditions of the Lagrange multipliers and evolve the system of ODE's until the terminal conditions given by Eqs 10 and 13 are met. We escape the numerical difficulties arising with this procedure by first guessing a solution and then finding the appropriate Lagrange multipliers

which verify optimality.

## Heuristic approach

Let us first consider what is necessary to keep the fraction of infected citizens at a constant value, $I_c$. Since $S$ varies with time, $dI/dt = 0$ entails $dI/dS = 0$, and hence from Eq 3 we obtain

$$\alpha(t) = 1 - \frac{1}{R_0 S(t)} \; . \tag{16}$$

This is indicated by the dash-dotted curve in Fig 2. Note that $\alpha(t)$ does not depend on the value of $I_c$.

Next we consider the cost function for proceeding from some $S = S_0$ to some $S = S_1 < S_0$ while maintaining $I = I_c$. Inserting Eq 16 in Eq 1, we find

$$dS = -I_c \frac{dt}{\tau} \; . \tag{17}$$

We use this substitution to express the cost function as

$$K = \int_{t(S_0)}^{t(S_1)} f(\alpha(t)) \, dt = \frac{\tau}{I_c} \int_{S_1}^{S_0} f(\alpha(S)) \, dS \, . \tag{18}$$

Hence if $S_0$ and $S_1$ are fixed, $I_c$ must be as large as possible to minimize $K$. This now guides our heuristic: we should control $\alpha$ such as to maintain $I = I_h$ for as long as possible.

If our guess is valid, the trajectory we have to follow in order to optimally control the pandemic is the one indicated as the solid curve in Fig 2. It starts at $(S, I) = (1, 0)$ and proceeds until $I = I_h$ is reached. This completes phase I of the process, during which we set $\alpha = 0$. Mitigation measures are then deployed, such that $\alpha$ jumps upwards to the dash-dotted curve. It follows that curve all through phase II, hence keeping $I = I_h$ constant. As $S$ decreases, $\alpha$ is gradually reduced until it reaches zero at the end of phase II. All through phase III, $\alpha$ is maintained at zero, while $I$ gradually decays to zero because $R < 1$. This ends the pandemic.

## Validation of the solution

We now proceed to verify our heuristic solution. We focus on phase II, as this is where the pandemic will spend the most amount of time. To do this we ask the question: Is it true that if the pandemic starts with $I_0 = I_h$, then for all $S_0 > 1/R_0$ and for all the cost functions $f(\alpha(t))$ such that $\frac{\partial f}{\partial \alpha}, \frac{\partial^2 f}{\partial \alpha^2} \geq 0$, optimal pandemic control must keep $I(t) = I_h$ until $S(t_e) = \frac{1}{R_0}$ is reached?

As we will see, the answer to the above question is yes. We proceed by setting $\alpha^*(t) = 1 - \frac{1}{R_0 S^*(t)}$ and $S^*(t) = S_0 - \frac{I_h}{\tau} t$ for $t \in [0, t_e^*]$ and $t_e^* = \left( S_0 - \frac{1}{R_0} \right) \frac{\tau}{I_0}$. The terminal conditions for the dynamics are given by $\boldsymbol{x}^*(t_e^*) = \left( \frac{1}{R_0}, I_h \right)$. We can substitute the terminal conditions in Eq 10 and get the terminal condition for $\lambda_S$ (the terminal condition for $\lambda_I$ is given by Eq 13). The task remains to find $\mu(t)$ such that Eqs 11 and 12 are satisfied simultaneously.

Let's have a look at Eqs 11 and 12 after making the substitutions. We have

$$\dot{\lambda}_S(t) = \frac{I_h}{\tau S^*(t)}\left[\lambda_S - \lambda_I\right],$$

$$\dot{\lambda}_I(t) = \frac{1}{\tau}\lambda_S + \mu(t),$$

(19)

and

$$\left.\frac{\partial f}{\partial \alpha}\right|_{\alpha^*(t)} = (\lambda_S(t) - \lambda_I(t))\beta I_h S^*(t).$$

(20)

One can hence find the Lagrange parameter $\mu(t)$ as

$$\mu(t) = -\frac{\lambda_S}{\tau} + \frac{\partial^2 f}{\partial \alpha^2}\frac{1}{R_0^2 S^{*3}}.$$

(21)

Lastly, there is also the issue of non-negativity of $\mu$. If we assume the convexity of $f$ we have $\frac{\partial^2 f}{\partial \alpha^2} \geq 0$. Hence the second summand in Eq 21 is non-negative. $\frac{\partial f}{\partial \alpha} \geq 0$ also implies that $\lambda_S$ is monotonically increasing (Eqs 19 and 20). If the cost of zero control is zero, then the terminal condition Eq 10 implies that $\lambda_S(t_e^*) = 0$, thereby implying $\lambda_S(t) \leq 0$ in the interval $[0, t_e^*]$. This shows that for time independent cost functions $f$ under the assumptions that $\frac{\partial f}{\partial \alpha}, \frac{\partial^2 f}{\partial \alpha^2} \geq 0$ and $f(0) = 0$ our heuristic solution is optimal in phase II.

## Numerical results

We have shown that an optimal trajectory starting on the boundary $(I_0 = I_h)$ remains on that boundary. To obtain optimal control trajectories for arbitrary initial conditions, we perform direct numerical optimization using the software library PSOPT [16]. In Fig 3 we show the numerical solutions to the control problem Eq (8) for various cost functions $f_i(\alpha(t)) \in \{\alpha(t), \alpha^2(t), \alpha^3(t)\}$.

Clearly, in all scenarios the optimal trajectory $I^*$ reaches the threshold value $I_h$ and remains there until $S^*(t_e^*)$ is reached (phase II). Phase I, however, depends on the cost function applied. For linear costs, $\alpha(t) = 0$ until $I = I_h$. With higher order cost terms, we observe non-zero control from the very beginning (see Fig 3). This is to reduce the amount of time spent at large control values $\alpha$ and thereby the total integrated costs. The optimal terminal time $t_e^*$ increases with the order of the cost function (see Fig 3). We should note, however, that the influence of the functional form of $f(\alpha(t))$, as expressed in the different shapes of the numerically derived curves, is minute, since the time axis is logarithmic, and the deviations are noticeable only during a very small fraction of time. Hence we see that the influence of the cost function, which corresponds to the chosen mitigation strategy, is finite, but can be regarded as *negligible* for practical purposes.

## Duration of the pandemic

If immune response acquired after recovery from an infection is permanent, the pandemic will last until herd immunity is reached at the end of phase II. This is when $S(t) = S_1 = 1/R_0$, as indicated by the left vertical dotted line in Fig 2. This is the start of phase III, in which the number of infected citizens decays with no mitigation measures anymore in place (i.e., at $\alpha = 0$). We will now discuss the time we expect it to take until this point is reached. Using Eq (17) with

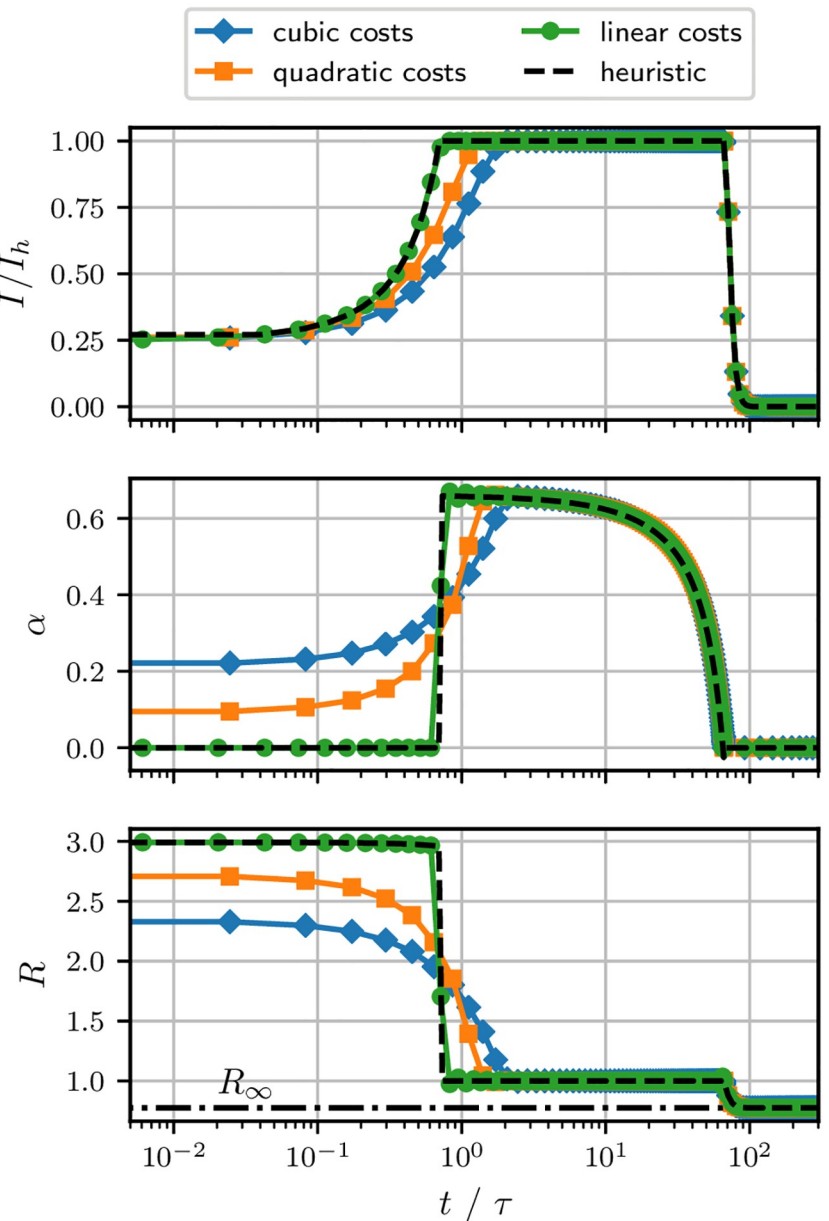

**Fig 3. Numerical solutions for optimal control.** Optimal control trajectories for different cost functions $f_i(\alpha(t)) \in \{\alpha(t), \alpha^2(t), \alpha^3(t)\}$. The corresponding optimal terminal times, $t_e^*$, are determined as $\{65.99\tau, 66.13\tau, 66.31\tau\}$. $I_0 = 0.0025$, $I_h = 0.01$, $R_0 = 3$, $S(t_e^*) = R_0^{-1}$. $R_\infty$ is the asymptotic reproduction number for $t \to \infty$, given by $R_\infty = -W(\exp(-1 - R_0 I_h))$, with the Lambert $W$ function.

$I_c = I_h$, we can write

$$dt = -\frac{\tau}{I_h} dS \tag{22}$$

and hence for the total duration of the pandemic, $T_0$, until herd immunity is reached,

$$T_0 = -\frac{\tau}{I_h} \int_{S_0}^{S_1} dS \approx \frac{\tau}{I_h} \left(1 - \frac{1}{R_0}\right). \tag{23}$$

Here we have exploited the fact that in all cases of practical relevance, $I_h$ will be very small as compared to unity. Consequently, the duration of phase I will be very small as compared to phase II, such that the evaluation of the true duration of phase I is of minor importance. As a very good approximation, we have simply set $S_0 = 1$ and neglected the impact of $\alpha(t)$ on the dynamics for the short period of phase I.

## Influence of immune response decay

The introductory discussion was based on the idea that recovered patients stay immune for all times. However, it is well known that for some diseases, in particular of the SARS-CoV type, the immune response tends to decay after some time [17]. Hence there is some finite probability that recovered patients become susceptible again.

We now assume that the transition from the recovered to the susceptible state can be described as a Poisson process. In other words, we assume the probability that a randomly chosen, formerly infected citizen becomes susceptible in a time interval $[t, t + dt]$ to be proportional to $dt$ and independent of $t$. This modifies the dynamical system (1) to

$$\partial_t S = -\beta SI + \frac{1}{\rho}(1 - S - I),$$

$$(24)$$

$$\partial_t I = \beta SI - \frac{I}{\tau},$$

with $\beta = \beta_0(1 - \alpha)$, and $\rho$ the average life time of the immune state, averaged over all formerly infected individuals. Note that we conceptually include those who fell victim to the disease and thus do not become susceptible again. Their contribution to the average resusceptibilization frequency is zero, which merely increases the average immune lifetime, $\rho$. From the data in [17], we find that after three years the average IgG immune response against SARS-CoV had decayed to 55.6 percent. For a corresponding Poissonian process we can estimate $\rho \approx 931$ days.

In Eq 24, we see immediately that the conditions to fulfill $\partial_t I = 0$ have not changed with respect to Eq 1. Hence the optimal control trajectory still obeys $\alpha(t) = 1 - \frac{1}{R_0 S(t)}$. In phase II, with optimal control, we obtain

$$\partial_t S = -\frac{I_h}{\tau} + \frac{1}{\rho}(1 - I_h) - \frac{1}{\rho}S \qquad (25)$$

with the solution

$$S(t) = I_h\left(1 + \frac{\rho}{\tau}\right)[e^{-t/\rho} - 1] + 1. \qquad (26)$$

Again, herd immunity (and hence the end of the pandemic) is reached when $S = 1/R_0$, at a time we call $T_r$, referring to resusceptibilization (i.e., decaying immune response). Inserting this into Eq 26 yields

$$1 - e^{-T_r/\rho} = \frac{1 - 1/R_0}{I_h(1 + \rho/\tau)} \qquad (27)$$

and hence

$$T_r = -\rho \ln\left[1 - \frac{1 - 1/R_0}{I_h(1 + \rho/\tau)}\right]. \qquad (28)$$

Although this looks rather awkward, it can be rewritten conveniently in terms of the pandemic duration, $T_0$, which we would find for infinite $\rho$. Defining the variable $X = T_0/(\tau + \rho)$, we find

$$\frac{T_r}{T_0} = -\frac{1}{X} \ln [1 - X].$$

(29)

$X$ is the total duration of the pandemic if no loss of immunity occurs, divided by the average time it takes for a patient from her infection to the loss of immunity after recovery, $\tau + \rho$. If immunity lasts very much longer than $T_0$, $X$ is small. In this case, the logarithm in Eq 29 can be expanded and we recover $T_r \approx T_0$. If, however, $\rho$ is of order $T_0$ (remember that $T_0 \gg \tau$ in all relevant cases), $T_r$ diverges. This behavior is summarized in Fig 4a, in which $X$ is chosen as the abscissa. We see that the duration of the pandemic becomes uncomfortably large when the total time from infection to resusceptibilization, $\tau + \rho$, comes close to the pandemic duration with infinite immunity, $T_0$ (vertical dotted line).

We might now ask how many acute infections the health system must be able to deal with in order to reach herd immunity at all. This can be derived by demanding $\lim_{t\to\infty} S(t) = 1/R_0$ in Eq 26. It is readily shown that the health system capacity required for reaching herd immunity is given by

$$\hat{I}_h = \frac{1 - \frac{1}{R_0}}{1 + \frac{\rho}{\tau}}.$$

(30)

This is shown in Fig 4b for different values of $R_0$. The dotted curve represents the (unrealistic) limiting case $R_0 \to \infty$.

Only with infinite immune response lifetime ($\rho \to \infty$), we observe an exponential decay of $I$ after herd immunity has been reached (see also Fig 3). To understand the long time dynamics after mitigation (phase III) for finite $\rho$, we draw the phase portrait (see Fig 5). There exist two fixed points, $(I_1, S_1) = (0, 1)$, a saddle for $R_0 \geq 1$, and $(I_\infty, S_\infty) = (\hat{I}_h, 1/R_0)$, a stable fixed point for $R_0 \geq 1$ (see S1 Appendix for details on the linear stability analysis). So for any initial conditions with $I_0 > 0$, the uncontrolled system will approach the stationary state, $(I_\infty, S_\infty)$. Interestingly, the stationary fraction of infected coincides with the minimal HSS capacity $\hat{I}_h$ needed to reach herd immunity.

We have thus shown that given $I_h > \hat{I}_h$, herd immunity can be reached in finite time during mitigation phase II. After mitigation measures have been released (phase III), $I$ converges to $\hat{I}_h$ in the long time limit. Moreover, $I$ remains below $I_h$ (see Fig 5) since re-entering the regime $I < I_h$ from above would require a trajectory to cross itself (not possible for an autonomous system of ODEs (Eq 24) with unique solutions).

## A few scenarios

Let us finally consider a few scenarios for some typical parameters, as we have in the current COVID-19 pandemic. The maximum number of known acute infections in Germany in spring 2020 was around 100.000, which was well tolerable for the HSS. Depending upon the percentage of cases which are officially recorded, the actual number of infected citizens may be considerably larger. If we assume a factor of two here, corresponding to 200.000 cases, we have $I_h \approx 0.0025$ (given $\approx$80 million citizens in Germany). On the other hand, if there are more, and if we also take into account that the HSS could well take a few more patients, we might also consider a scenario with 800.000 acute infections at a time, corresponding then to $I_h \approx$ 0.01. In both cases we also have to vary the average lifetime of the immune state, $\rho$, and observe its effect on the duration of the pandemic.

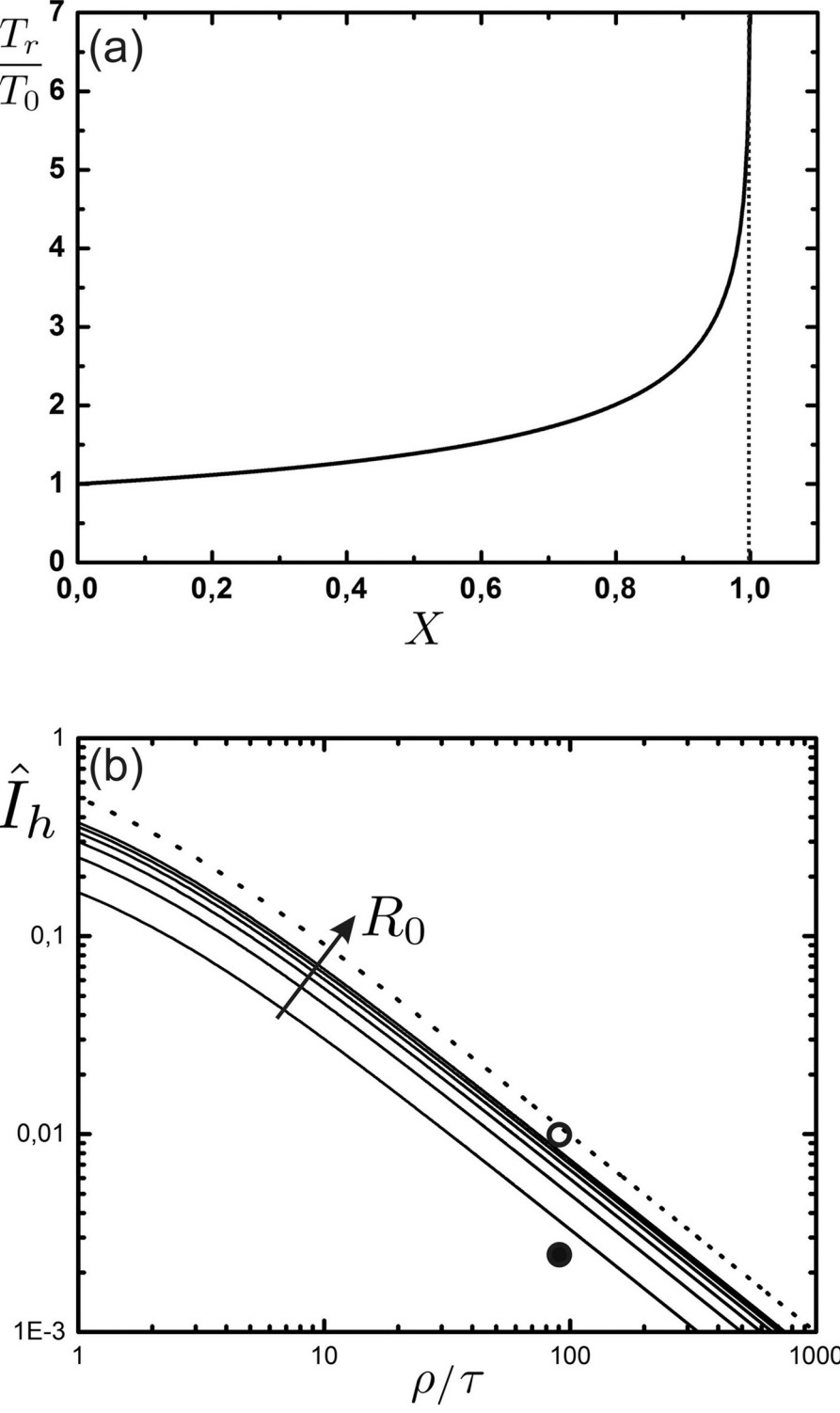

**Fig 4. Duration of the pandemic and minimum health system capacity.** (a) The normalized duration of the pandemic, $T_r/T_0$, as a function of the variable $X = T_0/(\tau + \rho)$ (Eq 29). (b) Solid curves: The minimum required health system capacity $\hat{I}_h$ to reach herd immunity (Eq 30) as a function of the duration of immunity after recovery, for different values of $R_0$ (from 1.5 to 4.0 in steps of 0.5). Dotted curve: limit $R_0 \to \infty$. Circles represent the scenario for $\rho = 93\tau$. Open: $I_h = 0.01$. Closed: $I_h = 0.0025$.

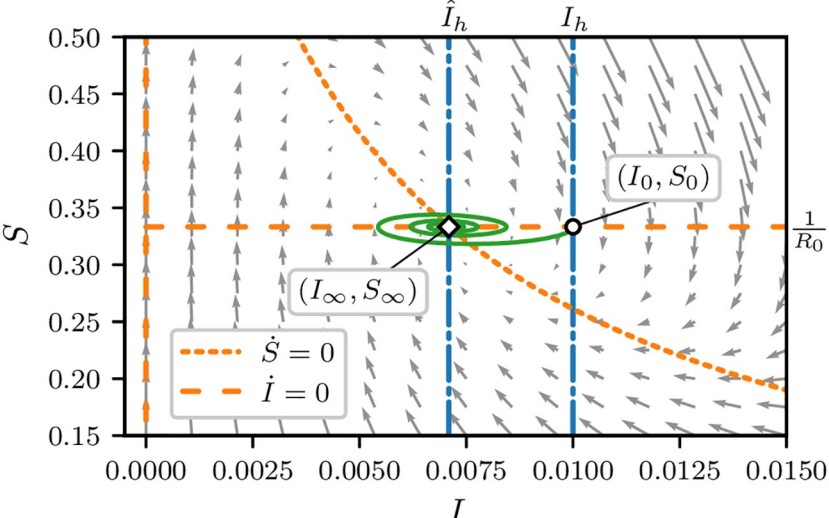

**Fig 5. Phase portrait of the uncontrolled SIR model.** Phase portrait, $(\dot{I}(S, I), \dot{S}(S, I))$, for the uncontrolled SIR model ($\alpha = 0$) with finite immune response (Eq 24). The solid green curve shows a trajectory in phase III, with initial conditions (circle) $I_0 = I_h$ (capacity limit) and $S_0 = 1/R_0$ (herd immunity), The dashed curves (orange) show the nullclines, $\dot{I} = 0$ (for $S = 1/R_0$ or $I = 0$), and $\dot{S} = 0$ (for $S = (1 - I)/(IR_0 \rho/\tau + 1)$). The stable fixed point (diamond) is given by $I_\infty = \hat{I}_h$, $S_\infty = 1/R_0$. Parameters: $R_0 = 3$, $\rho/\tau = 93$, $I_h = 0.01$.

The two sets of scenarios are represented in the graphs in Fig 6. The remaining fraction of susceptible citizens is shown as the solid curves, while the dashed curves represent the control parameter, $\alpha$. As before, we have assumed $R_0 = 3$ for the basic reproduction number. We take the value $\rho = 931$ days mentioned above for another SARS-CoV strain as a reasonable estimate. Using $\tau = $ ten days, this corresponds to $\rho = 93\tau$. In order to cover this case, we have used the values $\rho/\tau = \{50, 93, 200, \infty\}$. For Germany, an HSS capacity of $I_h = 0.01$ (top (a) graph) would correspond to roughly 32% of hospital beds. We do not consider the need for intensive care units here, one reason being lack of knowledge about the fraction of ICU cases. (500.000 in total) utilized for patients with COVID-19, if we assume a hospitalization rate of 20%. This is a conservative estimate given that the number published by the Robert Koch Institute [18] (17%) is based on reported cases only; the true hospitalization rate might be significantly lower. The bottom (b) graph corresponds to a smaller HSS capacity ($I_h = 0.0025$), for Germany, corresponding to 8% utilization of hospital beds.

From the steadily decreasing dashed curves representing $\alpha(t)$, it is obvious that the mitigation measures can be gradually alleviated as time proceeds. In the top (a) graph ($I_h = 0.01$) for infinite immunity ($\rho \to \infty$), one would reach the end of mitigation measures after about two years (= $66.7\tau$, with $\tau = 10$ days). This is, however, hardly realistic. For the more realistic case, $\rho/\tau = 93$, it would take about three years ($\approx 114.9\tau$). For an HSS capacity of $I_h = 0.0025$, bottom (b) graph, clearly, there would be no chance to ever reach herd immunity for $\rho/\tau = 93$. Instead, one would not reach any further than $\alpha \approx 0.5$, which still corresponds to rather harsh measures.

It should finally be noted that the number of fatalities is limited for all cases where herd immunity is reached. In particular, if $\phi$ is the fraction of fatalities among those which are infected, the fraction of fatalities with respect to the population is $F = \phi(1 - R_0^{-1})$ for infinite $\rho$. If $\rho$ is finite, we find

$$F = \frac{\phi T_r}{T_0}\left(1 - \frac{1}{R_0}\right) = \left(1 - \frac{1}{R_0}\right)\frac{\phi \ln(1 - X)}{X}. \tag{31}$$

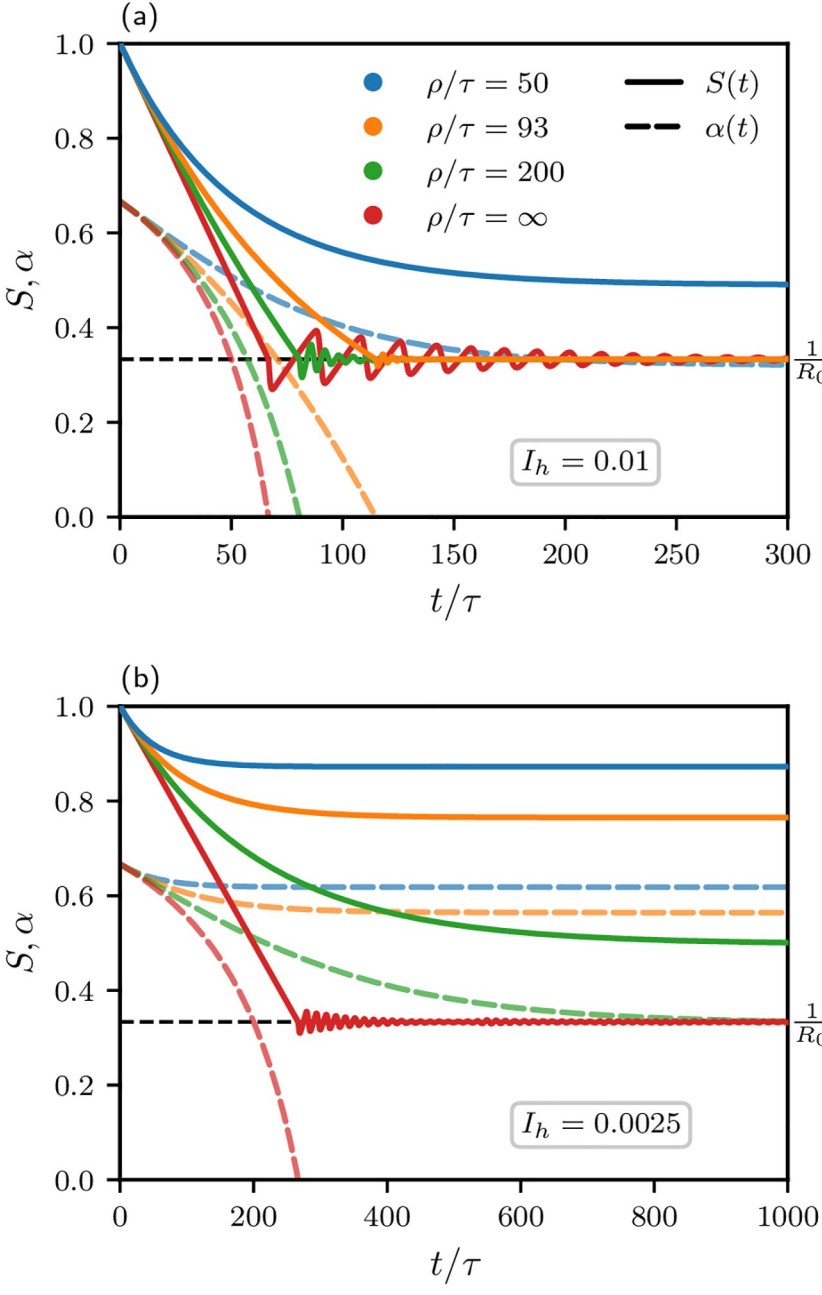

**Fig 6. Typical pandemic scenarios for different average immunity loss times, $\rho/\tau \in \{50, 93, 200, \infty\}$, corresponding to curves from right to left (or see color code), and different values for $I_h$, namely 0.01 in the top (a) graph and 0.0025 in the bottom (b) graph.** Solid curves: $S(t)$. Dashed curves: $\alpha(t)$. The fraction of acutely infected citizens is kept at $I_h$ in phase II until herd immunity is reached ($S = 1/R_0$, horizontal dashed line). If this is successful (if $I_h > \hat{I}_h$, see Eq 30) phase III begins: Mitigation measures are being released ($\alpha = 0$) and $S(t)$ oscillates around its limiting value $S_\infty = 1/R_0$. Other parameters: $R_0 = 3$, $\tau = 10$ days.

This is precisely the scaling described by the curve displayed in Fig 4a, which shows that already well below $X = 1$, the death toll incurred by the herd immunity strategy becomes prohibitively high if immune response decay plays a significant role.

## Conclusions

We have shown that for a wide class of cost functions, in order to reach herd immunity without vaccination, it provides an optimal control strategy to keep the effective reproduction number, $R$, at unity during the majority of the duration of the pandemic. Deviations which depend upon the specific form of the cost function are limited to a narrow time window and can be considered negligible for practical purposes.

Reducing $R$ can be achieved through various measures, e.g., increased hygiene, physical distancing, or contact tracing [19]. Keeping $R$ at the critical value of unity—above which epidemic spreading sets in—is, however, hardly feasible in practice, due to uncertainties as well as observation delays concerning the effects of mitigation measures. Development of robust optimal control scenarios taking such uncertainties into account is left for future investigations.

In this study, costs incurred at time $t$ have been considered local in time. Cost functions nonlocal in time (with a memory kernel) would be an interesting extension but go beyond the scope of this work. Costs associated with the number of infections have not been considered explicitly. Instead we kept the number of infections below an upper bound, e.g., the capacity limit of the health service system (HSS). Of course there are societal costs due to infections even below the limit of the HSS. However, if herd immunity is the goal and vaccination is not available, then there is no way around a fraction of $1 - 1/R_0$ of the population going through the infection. Moreover, the effectiveness of specific mitigation measures can depend on the number of infections; while contact tracing is an efficient measure for low case numbers, local health authorities can be overwhelmed if case numbers are high [20, 21]. Therefore, the socio-economic costs for establishing a given reproduction number $R$ might depend on $I$ as well. Here we focused on simple costs functions as a starting point allowing for analytical treatment.

Explicit expressions for the expected duration of the pandemic have been given, and we have seen that the duration of the pandemic increases strongly as the average lifetime of the immune state decreases. In particular, we can conclude that in case the immune response to SARS-CoV2 decays in a similar manner as for the formerly encountered SARS-CoV1 strain [17], using infection mediated herd immunity as a vaccination strategy for SARS-CoV2 would require a substantial fraction of health system capacity dedicated to COVID-19 patients (see Fig 6). However, as a consequence of global mobility there may be more pandemics coming which show different infection and immune response behavior. We therefore think that our results should be borne in mind for future use, as they are of rather general nature.

## Materials and methods

Simulation code and data can be found in the following repository: https://github.com/poss-group/covid19-control.

## Supporting information

**S1 Appendix. Detailed derivations.** 1) First order necessary conditions for optimality. 2) Stability analysis of the uncontrolled SIR model with finite immune response.
(PDF)

## Author Contributions

**Conceptualization:** Stephan Herminghaus, Knut M. Heidemann.

**Formal analysis:** Prakhar Godara, Stephan Herminghaus, Knut M. Heidemann.

**Investigation:** Prakhar Godara, Stephan Herminghaus, Knut M. Heidemann.

**Methodology:** Prakhar Godara, Stephan Herminghaus, Knut M. Heidemann.

**Software:** Knut M. Heidemann.

**Validation:** Knut M. Heidemann.

**Visualization:** Stephan Herminghaus, Knut M. Heidemann.

**Writing – original draft:** Prakhar Godara, Stephan Herminghaus, Knut M. Heidemann.

**Writing – review & editing:** Stephan Herminghaus, Knut M. Heidemann.

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
