## [Decision Letter · Decision Letter 0]

12 Jan 2021

PONE-D-20-30011

A control theory approach to optimal pandemic mitigation

PLOS ONE

Dear Dr. Heidemann,

Thank you for submitting your manuscript to PLOS ONE. After careful consideration, we feel that it has merit but does not fully meet PLOS ONE’s publication criteria as it currently stands. Therefore, we invite you to submit a revised version of the manuscript that addresses the points raised during the review process.

We look forward to receiving your revised manuscript.

Kind regards,

Alessandro Pluchino

Academic Editor

PLOS ONE

Journal Requirements:

"P. G., S. H., and K. M. H. gratefully acknowledge support from the Max Planck Society."

Reviewers' comments:

Reviewer's Responses to Questions

**Comments to the Author**

1. Is the manuscript technically sound, and do the data support the conclusions?

Reviewer #1: Partly

Reviewer #2: Yes

2. Has the statistical analysis been performed appropriately and rigorously? 

Reviewer #1: N/A

Reviewer #2: Yes

3. Have the authors made all data underlying the findings in their manuscript fully available?

Reviewer #1: Yes

Reviewer #2: Yes

4. Is the manuscript presented in an intelligible fashion and written in standard English?

Reviewer #1: Yes

Reviewer #2: Yes

5. Review Comments to the Author

Reviewer #1: In effect, the authors examine the incurring costs of Corona when

the level of infections is kept at the boundary of the medical capabilities.

This is a valid investigation which should be published. There are however

a few points.

(a) Figures are of --very-- low quality (resolution). Figure captions are highly

insufficient. For example, in Fig. 1 the cost function f(alpha) is mentioned as

'defined somewhere in the text'. Suggesting that the reader should take the

trouble to search for her/him-self. That is arrogant.

(b) Herd immunity is taken at the endpoint. In reality, the outbreak will take

substantially longer, unit I(t)=0. There are not estimates about the costs

incurring after the herd immunity point. Without this estimate, the paper cannot

be published.

(c) The incurring costs are taken to instantaneous. In reality, a certain level

of containment may be ok for a few weeks, but disastrous if uphold for many

years, as suggested by the authors. For example, for a few weeks/month

business may close and reopen, but not after many years. The authors should

clearly state that they propose a duration of 4-6 years (you need to include the

post-here period!) for Germany. The issue of additional costs incurring for

prolong lockdowns need to be discussed.

Reviewer #2: This manuscript introduce a control theory approach to identify optimal pandemic mitigation strategies based on the framework of homogeneous susceptible-infected-recovered (SIR) models. It provides an optimal control strategy. Optimality is derived and verified by variational and numerical methods for a number of model cost functions. And also they discussed in terms of the feasibility of strategies based on herd immunity. It has certain theoretical guiding significance.

6. PLOS authors have the option to publish the peer review history of their article (what does this mean?). If published, this will include your full peer review and any attached files.

Reviewer #1: No

Reviewer #2: No

---

## [Author Response · Author response to Decision Letter 0]

5 Feb 2021

Dear Prof. Pluchino, Dear Reviewers,

We highly appreciate your comments regarding our manuscript. Please find our detailed response in the separate file "Letter to the Reviewers".

Kind regards,

–Knut Heidemann.

---

## [Editor Report · Decision Letter 1]

8 Feb 2021

A control theory approach to optimal pandemic mitigation

PONE-D-20-30011R1

Dear Dr. Heidemann,

We’re pleased to inform you that your manuscript has been judged scientifically suitable for publication and will be formally accepted for publication once it meets all outstanding technical requirements.

Kind regards,

Alessandro Pluchino

Academic Editor

PLOS ONE
---

## [Editor Report · Acceptance letter]

11 Feb 2021

PONE-D-20-30011R1 

A control theory approach to optimal pandemic mitigation 

Dear Dr. Heidemann:

I'm pleased to inform you that your manuscript has been deemed suitable for publication in PLOS ONE. Congratulations! Your manuscript is now with our production department. 

Kind regards, 

on behalf of

Dr. Alessandro Pluchino 

Academic Editor

PLOS ONE